

# The effects of trunk kinematics and EMG activity of wheelchair racing T54 athletes on wheelchair propulsion speeds

Wei Guo[1,2], Qian Liu[3], Peng Huang[4], Dan Wang[1], Lin Shi[5] and Dong Han[1]

[1] School of Athletic Performance, Shanghai University of Sport, Shanghai, China
[2] Shaanxi XueQian Normal University, Xi'an, China
[3] Key Laboratory of Exercise and Health Sciences of Ministry of Education, Shanghai University of Sport, Shanghai, China
[4] Shanghai Culture and Sports Promotion Center for Persons with Disabilities, Shanghai, China
[5] School of Physical Education, Chengdu Sport University, Chengdu, China

Corresponding author
Dong Han, handtiyu@126.com

## ABSTRACT

**Background**. The purpose of this study is to examine the impact of trunk kinematic characteristics and trunk muscle electromyography (EMG) activity on propulsion speeds in wheelchair racing T54 athletes.

**Method**. The Vicon infrared high-speed 3D motion capture system was utilized to acquire kinematic data of the shoulders, elbows, wrists, and trunk from twelve T54 athletes at four different speeds (5.55 m/s, 6.94 m/s, 8.33 m/s, and personal maximum speed). Additionally, the Trigno Wireless EMG system was employed to collect synchronous surface electromyography (EMG) data from the rectus abdominis and erector spinae muscles. The kinematics and EMG data of the trunk were compared across various wheelchair propulsion speeds while also examining the correlation coefficient between wheelchair propulsion speeds and: (1) the range of motion of upper limb joints as well as the trunk; (2) the maximum angular velocities of the upper limbs joints as well as the trunk; and (3) rectus abdominis and erector spinae EMG activity. Two multiple linear stepwise regression models were utilized to examine the impact of variables that had been identified as significant through correlation coefficient tests (1) and (2) on propulsion speed, respectively.

**Results**. There were significant differences in the range of motion ($p<0.01$) and angular velocity ($p<0.01$) of the athlete's trunk between different propulsion speeds. The range of motion ($p<0.01$, $r = 0.725$) and angular speed ($p<0.01$, $r = 0.882$) of the trunk showed a stronger correlation with propulsion speed than did upper limb joint movements. The multiple linear stepwise regression model revealed that the standardized $\beta$ values of trunk motion range and angular velocity in athletes were greater than those of other independent variables in both models. In terms of the EMG variables, four of six variables from the rectus abdominis showed differences at different speeds ($p<0.01$), one of six variables from the erector spinae showed differences at different speeds ($p<0.01$). All six variables derived from the rectus abdominis exhibited a significant correlation with propulsion speed ($p<0.05$, $r>0.3$), while one variable derived from the erector spinae was found to be significantly correlated with propulsion speed ($p<0.01$, $r = 0.551$).

**Conclusion**. The movement of the trunk plays a pivotal role in determining the propulsion speed of wheelchair racing T54 athletes. Athletes are advised to utilize trunk

movements to enhance their wheelchair's propulsion speed while also being mindful of the potential negative impact on sports performance resulting from excessive trunk elevation. The findings of this study indicate that it would be beneficial for wheelchair racing T54 athletes to incorporate trunk strength training into their overall strength training regimen, with a specific emphasis on enhancing the flexion and extension muscles of the trunk.

# INTRODUCTION

The athlete's trunk position and movement exert a significant impact on the propulsion technique (*Lewis et al., 2019*; *Moss, Fowler & Goosey-Tolfrey, 2005*; *Ridgway, Wilkerson & Pope, 1987*; *Sanderson & Sommer, 1985*) and propulsion speed (*Gehlsen, Davis & Bahamonde, 1990*) of wheelchair racing athletes, as it is one of the primary means to generate propulsive force (*Vanlandewijck, Theisen & Daly, 2001*). The first published article on the technical aspects of wheelchair racing shows that trunk movements significantly impact the force that is transferred from the trunk to the handrim and that the maximum force application point is on the handrim (*Sanderson & Sommer, 1985*). The flexion of the trunk during wheelchair propulsion creates a favorable position for the upper limbs and hands to exert force on the handrim (*Moss, Fowler & Goosey-Tolfrey, 2005*). This is attributed to the inclined position of the trunk, which increases the angle of hand contact with the handrim (*Gehlsen, Davis & Bahamonde, 1990*; *Wang et al., 1995*) and generates a significant amount of vertical motion (*Wang et al., 1995*) during propulsion and expands the range of motion throughout the entire propulsive phase (*Chow et al., 2001*; *Gehlsen, Davis & Bahamonde, 1990*), thereby contributing to acceleration (*Wang et al., 1995*). In situations where athletes require greater force generation for propulsion, they not only utilize arm movements but also lean their trunk forward to generate additional momentum at the handrim (*Wang et al., 1995*). Additionally, the movement and strength of the athlete's trunk are extremely crucial for starting the wheelchair from a stationary state (*Lewis et al., 2019*; *Vanlandewijck, Theisen & Daly, 2001*). This is due to the fact that peak velocity during sprint starts in wheelchair racing is directly linked to the flexion and extension of the athlete's trunk (*Moss, Fowler & Goosey-Tolfrey, 2005*). Therefore, the trunk movement of T54 wheelchair racing athletes is a critical determinant of their athletic performance.

The technical economy (*Forte et al., 2019a*; *Goosey & Campbell, 1998a*; *Goosey, Campbell & Fowler, 1998*; *Jones et al., 1992*; *Vanlandewijck, Spaepen & Lysens, 1994*) and air drag (*Barbosa et al., 2016*; *Forte et al., 2018a*; *Hedrick et al., 1990*; *Lewis et al., 2017*) of wheelchair athletes are affected by their trunk posture and movements. Athletes who possess superior technical efficiency demonstrate enhanced rhythm perception and reduced trunk movement velocities during propulsion (*Jones et al., 1992*). A study on the technical economy of wheelchair basketball players' propulsion revealed a correlation between

reduced technical efficiency and increased trunk motion (*Vanlandewijck, Spaepen & Lysens, 1994*). Moreover, the position of the trunk has a significant impact on the air resistance encountered by athletes during propulsion (*Barbosa et al., 2016*), with an increase in air drag observed when the trunk is elevated (*Akashi et al., 2019*). By adopting a more aerodynamic posture or utilizing trunk flexion, athletes can reduce this area of their trunk by 0.17 m$^2$ (*Barbosa et al., 2016*) or by 18% (*Hedrick et al., 1990*) respectively, thereby minimizing air drag. If male wheelchair racing athletes were to adopt these techniques, they could potentially improve their 5,000 m race times by up to 116 seconds (*Lewis et al., 2017*). By optimizing trunk posture and range of movement, wheelchair racing athletes can enhance their technical efficiency and reduce air resistance, ultimately improving their competitive performance.

Elite male athletes exhibit a greater range of trunk movement compared to their younger (*Goosey, Fowler & Campbell, 1997*) and female (*Lewis et al., 2017*) counterparts. Furthermore, the trunk movements of wheelchair racing athletes differ based on their classification levels (*Gehlsen, Davis & Bahamonde, 1990*; *Goosey-Tolfrey et al., 2001*; *Ridgway, Pope & Wilkerson, 1988*). Athletes with limited or no trunk mobility face significant challenges in achieving optimal sports performance when compared to those with full trunk mobility (*Connick et al., 2018*). Based on the findings of these studies, it can be speculated that trunk movement may constitute a significant factor elucidating the disparate competition outcomes across genders, ages, and disability categories. Notably, T54 is the sole category featuring complete trunk functionality among wheelchair racing T51–T54 athletes (*Vanlandewijck et al., 2011*). Nevertheless, to date, few studies have been conducted on the kinematics and electromyography (EMG) activity of the trunk in wheelchair racing T54 athletes.

Only a limited number of studies have utilized electromyography (EMG) to examine muscle activation during wheelchair propulsion among athletes. *Chow et al. (2001)* employed EMG to investigate the contraction characteristics of upper limb muscles in wheelchair athletes utilizing two distinct racing techniques and varying levels of resistance (*Chow et al., 2000*). A study has investigated the disparities in EMG characteristics of the triceps muscle between elite and amateur wheelchair marathon runners (*Umezu et al., 2003*). While these studies focused on upper-limb EMG features in wheelchair athletes, they did not include data on trunk EMG. Currently, only one study has been conducted on the EMG characteristics of the trunk in T54 wheelchair racing athletes (*Kumnerddee et al., 2018*), which demonstrated that abdominal function was most activated and associated with propulsion speed. Moreover, elite athletes exhibit a proclivity towards utilizing their rectus abdominis to a greater extent than their slower counterparts (*Kumnerddee et al., 2018*). A study examining daily wheelchair propulsion *via* trunk EMG revealed that the simultaneous activation of both the abdominal and back muscles during the initial stage of movement enhanced handrim force efficiency (*Jones et al., 1992*), with an increase in trunk muscle activation observed as propulsion speed escalated. These two studies are significant contributions to the research on the technical characteristics of wheelchair racing events, indicating that the activation level of the rectus abdominis and erector spinae muscles during propulsion may potentially impact performance in wheelchair racing T54 athletes.
To date, the majority of studies on kinematic research in wheelchair racing athletes have primarily focused on timing parameters (the proportion of the propulsion phase to the recovery phase in a stroke cycle, etc.) (*Chow & Chae, 2007*; *Chow et al., 2001*; *Goosey & Campbell, 1998b*; *Goosey, Fowler & Campbell, 1997*; *Moss, Fowler & Goosey-Tolfrey, 2005*; *O'Connor, Robertson & Cooper, 1998*; *Ridgway, Pope & Wilkerson, 1988*; *Sanderson & Sommer, 1985*; *Wang et al., 1995*) and upper limb movements (*Chow et al., 2000*; *Chow et al., 2001*; *Goosey, Fowler & Campbell, 1997*; *Ridgway, Pope & Wilkerson, 1988*; *Ridgway, Wilkerson & Pope, 1987*; *Sanderson & Sommer, 1985*; *Wang et al., 1995*; *Wang, Vrongistinos & Xu, 2008*). Despite some previous studies on the trunk kinematics of wheelchair racing athletes (*Kumnerddee et al., 2018*; *Lewis et al., 2017*), no research has yet investigated the trunk movement of T54 wheelchair racers at different speeds using both 3D motion capture and EMG. The purpose of this study was to examine the impact of the kinematics and EMG characteristics of the trunk in T54 wheelchair racers on propulsion speeds. We hypothesized that the movement of the trunk in T54 wheelchair racing athletes would exert a significant impact on wheelchair propulsion speed, and that rectus abdominis and erector spinae muscle EMG activity would escalate with increased propulsion speed.

## METHODS

### Participants

This study included active wheelchair racing athletes at the T54 wheelchair racing level who had no prior injuries and refrained from using drugs or alcohol before the trial. A total of twelve athletes were registered, comprising ten male and two female competitors, six of whom were members of the Chinese national team with prior experience in international competitions (Paralympic Games, Asian Paralympic Games), and six other athletes who had competed in national-level events (National Paralympic Games, National Championships). The athletes underwent training for a period ranging from four to 14 years, with eight of them adopting the kneeling posture—four of whom had polio, three suffered from SCI (spinal cord injury), and one was an amputee. Meanwhile, four athletes were seated—three of whom were amputees and one had polio. All athletes provided written informed consent prior to participating in the experiment, and this study was approved by the Ethics Committee of Shanghai University of Sport (IRB approval number: 102772021RT104). Table 1 displays the physical characteristics of the athletes included in this investigation.

### Instrumentation

The kinematic data were acquired through the utilization of a Vicon infrared High-speed Motion Capture system (T40), consisting of 10 cameras (VICON Motion Systems, Oxford, UK) with a sampling frequency of 200 Hz. A total of 36 retro-reflective markers, measuring 14 mm in diameter, were affixed to the bone landmarks located on the trunk and upper limb regions to define the shoulder, elbow, wrist, and trunk.

The Trigno Wireless EMG System (Delsys, Natick, MA, USA), consisting of a 16-wire EMG test system with a wireless sensor (EMG signal width: 20-450 Hz; signal sample rate: 2,000 sample/s) and a base station, was utilized for surface EMG data acquisition in this study. A sampling frequency of 2,000 Hz was employed. The rectus abdominis

**Table 1** Physical characteristics of the wheelchair racing T54 athletes included in the study.

|  | n | Age (yr) | Sitting Height (cm) | Weight (kg) | Years for training (yr) |
|---|---|---|---|---|---|
| Man | 10 | 21.7 ± 4.22 | 89.7 ± 6.13 | 79.9 ± 9.68 | 6.7 ± 2.87 |
| Woman | 2 | 28 ± 5.66 | 83.5 ± 0.71 | 72.3 ± 6.9 | 9 ± 7.07 |
| Total | 12 | 22.75 ± 4.85 | 88.67 ± 6.05 | 76.3 ± 9.23 | 7.08 ± 3.48 |

and erector spinae were selected as the electrode placement sites based on the SENIAM recommendations for sensor locations (*Hermens et al., 1999*).

## Procedure

Prior to testing, each participant engaged in a warm-up routine consisting of stretching their muscles and joints, including the head, neck, shoulder, elbow, and trunk. They then proceeded to warm up on their own unmodified racing wheelchair, which they used in daily training and competition, at moderate speeds determined by individual preference. The wheelchairs were secured onto a training roller (D&J, USA), and all test instruments were checked during this 5-minute period. A speedometer mounted on the racing wheelchair was used to monitor wheelchair propulsion speed during the experiment.

## Data collection

The athlete started propulsion when instructed by the operator, as depicted in Fig. 1. The athlete executed 5 s of propulsion at each speed, and three complete push cycle records were obtained for analysis at each speed (*Chow et al., 2001*; *Sanderson & Sommer, 1985*; *Wang et al., 1995*). Athletes took 5-minute breaks between each speed to prevent fatigue. A total of four speed tests were conducted: 5.55 m/s (20 km/h), 6.94 m/s (25 km/h), 8.33 m/s (30 km/h) and personal maximum speed, which are comparable to races and training sessions.

The Trigno Wireless EMG telemetry system and the VICON system were utilized for synchronous testing, with the former collecting EMG data and the latter kinematic data from athletes. The maximum voluntary contraction (MVC) test was conducted without fatigue (*Rejc et al., 2010*). MVC values were obtained by performing maximal isometric contractions of the rectus abdominis and erector spinae on a bench for 5 s, repeated three times, with the highest value being recorded (*Beierle et al., 2019*).

## Data processing

The Nexus signal acquisition and processing software, developed by VICON (VICON Nexus 2.6.1), was utilized for the collection of kinematic signals with marker naming, noise removal, track deletion, and other early signal processing procedures conducted post-data collection (*Coker et al., 2021*). Kinematic data collected in this study were calculated using Visual3D analysis software (V3D, Version 6, C-Motion Inc., Germantown, MD, USA).

The EMG data collected was bandpass filtered using the EMG Works 4.5 analysis software (DELSYS Inc., Natick, MA, USA) with a Butterworth filter having a passband width of 10-393 Hz. The software performed baseline adjustment by removing the mean, full wave rectification, and wave rectification to 1,000 Hz. The final EMG data was exported

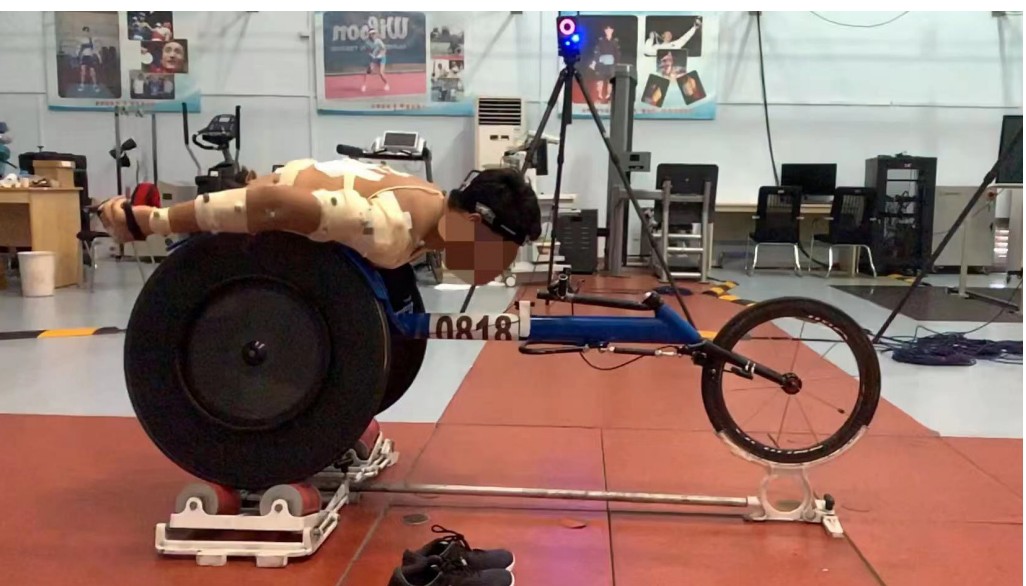

**Figure 1** The scene of the technical test for wheelchair racing athletes.

to a C3D file and synchronized with kinematic data in Visual3D (Version 6, V3D) software (*Visual3D, 2023*).

Based on previous research, this study has categorized the technical movements of athletes into three phases (*Chow et al., 2001*; *Cooper, 1990*; *Forte, Barbosa & Marinho, 2015*; *Forte et al., 2018b*): propulsion, release, and recovery, as well as three specific time points-hand contact with the wheel, hand off the wheel, and the highest point of elbow elevation. The propulsive phase commences upon handrim contact and persists until release. The release phase initiates as the athlete's hand disengages from the handrim and endures until the elbow reaches its highest point. The recovery phase denotes the interval between elbow highest point and subsequent handrim contact. In kinematic data, the $X$-axis represents joint flexion or extension in the sagittal plane, while the $Y$-axis represents joint abduction or adduction in the coronal plane. The $Z$-axis denotes external or internal rotation of the joint in the horizontal plane. Additionally, lean angle indicates maximum trunk downward inclination, and raised angle signifies maximum trunk elevation, both angles are measured within the sagittal plane.

## Statistical analysis

The statistical analysis in this study was conducted using IBM Statistical Package (IBM SPSS Inc., Chicago, IL, USA). Prior to the analysis, the data for each group underwent the Shapiro–Wilk normality test. The mean ± standard deviation was used to express the angle, range of motion, and angular velocity of the trunk at different speeds due to their normal distribution. Meanwhile, median and interquartile range were utilized to express EMG data of the rectus abdominis and erector spinae at different speeds due to their abnormal distribution. Therefore, parametric tests are utilized for statistical analysis of kinematic data, and non-parametric tests are utilized for statistical analysis of EMG data,

respectively. One-way ANOVA was utilized to compare the trunk motion angle and angular velocity among different speeds, while the Kruskal-Wallis test was employed to compare EMG variables of the rectus abdominis and erector spinae among different speeds. Partial eta-squared ($\eta 2$) was calculated as the effect size to evaluate the significance of significant findings for one-way ANOVA and Kruskal-Wallis. The Pearson correlation coefficient test was utilized to examine the relationship between propulsion speed and (1) each joint's range of motion in the X, Y, and Z axes at various propulsion speeds; (2) the maximum angular velocity of each joint in the X, Y, and Z axes at various propulsion speeds. Variables that showed a significant correlation with propulsion speed ($p < 0.05$) in both Pearson correlation coefficient tests were utilized to construct separate multiple stepwise regression models for each group, respectively. The Spearman correlation coefficient test was utilized to investigate the relationships between wheelchair propulsion speeds and EMG activity of the athlete's rectus abdominis and erector spinae, respectively. Significance for all tests was assumed at $p < 0.05$.

## RESULTS

The results of the one-way ANOVA test indicate (Table 2 and Fig. 2) significant differences among different speeds in terms of raised angle ($p < 0.05$, $\eta 2 = 0.164$), range of movement ($p < 0.001$, $\eta 2 = 0.573$), and angular velocity of the trunk ($p < 0.001$, $\eta 2 = 0.796$). However, there were no significant differences observed in lean angle across different speeds ($p > 0.05$, $\eta 2 = 0.044$).

Fisher's least significant difference (LSD) *post-hoc* tests revealed significant differences in the raised angles between maximum speed and both 5.55 m/s and 6.94 m/s ($p < 0.01$, $p < 0.05$). Moreover, there were highly significant differences in the range of motion between maximum speed and the other three speeds ($p < 0.01$, $p < 0.01$, $p < 0.01$), as well as between 5.55 m/s and 8.33 m/s ($p < 0.01$). There were also significant differences in angular velocity between the maximum speed and the other three speeds ($p < 0.001$, $p < 0.001$, $p < 0.001$), as well as between 5.55 m/s and 8.33 m/s ($p > 0.01$).

The correlation analysis between the range of motion and propulsion speed revealed that both the trunk (X-axis, $p > 0.01$, $r = 0.725$) and shoulder (X-axis, left: $p > 0.01$, $r = 0.624$; right: $p > 0.01$, $r = 0.642$) were significantly associated with propulsion speed, with correlation coefficients exceeding 0.6. Among all variables tested, the trunk on the X axis exhibited the highest correlation coefficient. The left shoulder on the Y-axis ($p < 0.05$, $r = 0.285$), as well as the left and right shoulder joints on the Z-axis (left: $p < 0.05$, $r = 0.326$; right side: $p < 0.01$, $r = 0.39$), exhibited significant correlations with propulsion speed, while no other variables were found to be significantly correlated.

The results of the correlation analysis indicate that all variables, except for the left wrist on the Z-axis ($p = 0.496$, $r = 0.101$), exhibited a significant correlation with propulsion speed ($p < 0.05$, $r > 0.3$) when compared to the maximum angular velocity of an athlete's trunk, shoulder, elbow, and wrist on the X, Y, and Z axes. Among these variables, the

**Table 2  The ANOVA results for trunk angle and angular speed (X-axis) at different propulsion speeds.**

| Speed (m/s) | Lean angle (degrees) | Raised angle (degrees) | Rang of motion (degrees) | Angular velocity (degrees/s) |
|---|---|---|---|---|
| 5.55 | 98.019 (8.190) | 87.003 (9.837)[**] | 11.02 (4.47)[*****] | 51.99 (18.04)[*****] |
| 6.94 | 99.094 (8.699) | 85.290 (9.927)[**] | 13.8 (5.5)[**] | 68.71 (24.01)[****] |
| 8.33 | 99.970 (8.687) | 82.069 (10.130) | 17.9 (6.48)[**] | 91.38 (31.08)[**] |
| Max | 102.653(7.803) | 76.325(8.417) | 26.33(4.05) | 180(29.37) |
| **p** | **0.572** | **0.047** | **0.000** | **0.000** |
| $\eta^2$ | **0.044** | **0.164** | **0.573** | **0.796** |

Notes.

\* and ** different from maximum speed for a $p < 0.05$ and $<0.01$ (respectively)

*** and **** different from speed at 8.33 m/s for a $p < 0.05$ and $<0.01$ (respectively). Standard deviations are presented in parenthesis. The bold values indicate the statistical results.

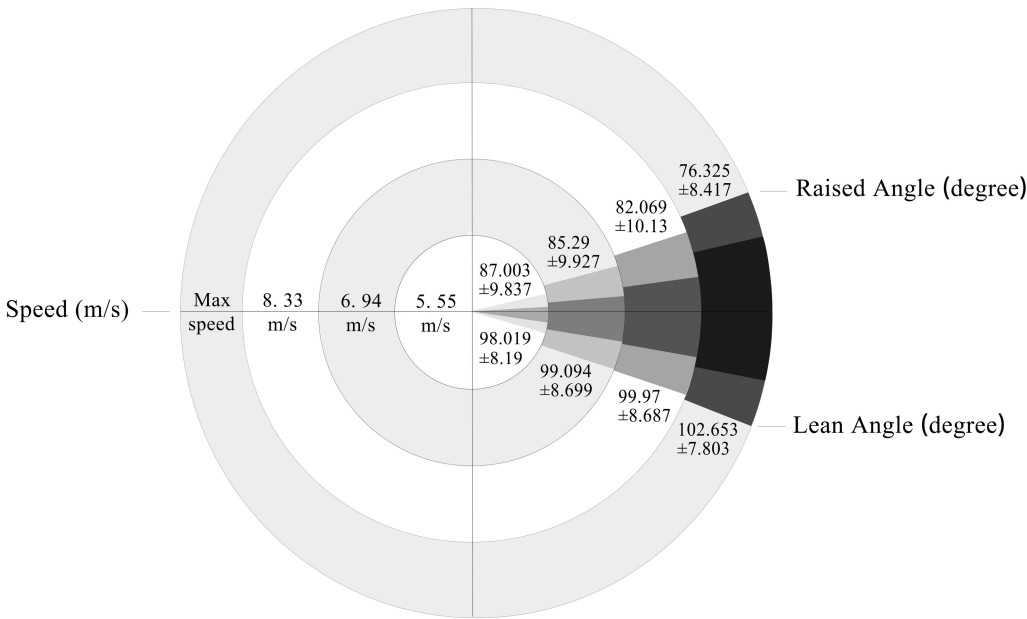

**Figure 2  The range of movement of athlete's trunk at different speeds.**

maximum angular velocity of an athlete's trunk on the $X$-axis was found to be most significantly correlated with propulsion speed ($r = 0.882$, $p > 0.01$).

Based on the aforementioned results, multiple linear stepwise regression analyses were conducted to examine the correlation between propulsion speed and range of motion in the $X$-axis of the trunk, the $Y$-axis of the left shoulder, the X and Z-axes of both shoulders, as well as maximum angular joint velocity in the X, Y, and Z-axes (excluding the $Z$-axis for the left wrist) of the trunk, shoulder, elbow, and wrist. The results are presented in Table 3.

The range of motion of the trunk on the X axis (X-Trunk), right shoulder on the X axis (X-RSHO), and left shoulder on the Y axis (Y-LSHO) were included in a multiple linear stepwise regression model (Table 3). The overall regression model was statistically

**Table 3  Multiple stepwise regression of propulsion speed to maximum angular joint velocity and to joint range of motion.**

| | | $\beta$ | t | p | VIF | $\Delta R^2$ | p |
|---|---|---|---|---|---|---|---|
| *Joint motion range* | X-Trunk | 0.484 | 3.490 | 0.001 | 2.287 | | |
| | X-RSHO | 0.307 | 2.205 | 0.033 | 2.298 | 0.605 | $p < 0.01$ |
| | Y-LSHO | 0.288 | 3.082 | 0.004 | 1.035 | | |
| *Maximum angular joint velocity* | X-Trunk | 0.422 | 4.070 | 0.000 | 3.944 | | |
| | X-LSHO | 0.228 | 2.119 | 0.040 | 4.253 | | |
| | Z-LELB | 0.167 | 2.810 | 0.007 | 1.305 | 0.872 | $p < 0.01$ |
| | Y-RSHO | 0.164 | 2.422 | 0.020 | 1.689 | | |
| | X-RSHO | 0.163 | 1.708 | 0.095 | 3.334 | | |

Notes.
Dependent Variable: propulsion speed

significant ($p < 0.001$) and accounted for 60.5% of the variance in wheelchair propulsion speed ($\Delta R^2 = 0.605$). The trunk exhibited the highest $\beta$ value along the $X$-axis (0.484), followed by the right shoulder joint with a $\beta$ value of 0.307, and finally, the left shoulder joint had the lowest $\beta$ value along the $Y$-axis (0.288).

The regression model incorporated the maximum angular velocities of the X-Trunk, X-RSHO, and X-LSHO joints on the X axis, the Y-RSHO joint on the Y axis, and the Z-LELB joint on the Z axis. It accounted for 87.2% of wheelchair propulsion speed ($\Delta R^2 = 0.872$). Among all variables included in this model, the $\beta$ value was highest for the maximum angular velocity of the trunk on the $X$-axis ($\beta = 0.422$).

The EMG signals of the rectus abdominis and erector spinae were recorded at specific time points during the task, including: onset of hand contact with the handrim (P-RMS), propulsive phase (P-Int), release of hand from the handrim (R-RMS), release phase (R-Int), highest point of elbow flexion during recovery (RE-RMS), and recovery phase (RE-Int).

All data in Table 4 were expressed as median and interquartile range due to an abnormal distribution. The Kruskal-Wallis test results indicated significant differences among different speeds in P-RMS ($p > 0.01$, $\eta 2 = 0.275$), RE-RMS ($p > 0.01$, $\eta 2 = 0.496$), R-Int ($p > 0.01$, $\eta 2 = 0.313$) and RE-Int ($p > 0.01$, $\eta 2 = 0.307$) of the rectus abdominis, as well as R-RMS of the erector spinae ($p > 0.01$, $\eta 2 = 0.346$).

Table 5 displays the correlation coefficients between the root mean square (RMS), the integrated EMG of the trunk, and propulsion speed. The rectus abdominis' EMG was correlated with propulsion speed throughout all time periods, with the highest correlation coefficient observed during the recovery phase (RE-Int, $p > 0.01$, $r = 0.714$). The correlation between erector spinae muscles and wheelchair propulsion speed was found to be significant solely when the hand leaves the wheel (R-RMS, $p > 0.01$, $r = 0.551$), with no statistical significance observed during other time periods.

## DISCUSSION

This study investigated the impact of trunk movement on wheelchair propulsion speed by analyzing the range of motion, maximum angular velocity of the trunk, and EMG activity of trunk muscles in T54 wheelchair racing athletes at various propulsion speeds.

Table 4 The integrated electromyography (iEMG), median and interquartile range (IQR) of rectus abdominis and erector spinae at different propulsion speeds.

| | Speed (m/s) | P-RMS | R-RMS | RE-RMS | P-Int | R-Int | RE-Int |
|---|---|---|---|---|---|---|---|
| Rectus abdominis | 5.55 | 0.02 (0.11)[**] | 0.00 (0.08) | 0.01 (0.10)[**] | 5.65 (46.39) | 0.94 (93.20)[**] | 2.92 (14.32)[**] |
| | 6.94 | 0.16 (0.27) | 0.00 (0.11) | 0.08 (0.22)[**] | 21.45 (57.66) | 29.49 (92.00)[*] | 24.88 (50.40)[**] |
| | 8.33 | 0.22 (0.50) | 0.01 (0.19) | 0.17 (0.30) | 46.98 (94.08) | 56.65 (136.20) | 45.96 (81.30) |
| | Max | 0.35 (0.73) | 0.27 (1.07) | 0.47 (0.48) | 48.32 (177.14) | 173.26 (83.58) | 120.99 (111.39) |
| | p | 0.001 | 0.050 | 0.000 | 0.119 | 0.002 | 0.000 |
| | $\eta^2$ | 0.275 | 0.283 | 0.496 | 0.046 | 0.313 | 0.307 |
| Erector spinae | 5.55 | 0.06 (0.19) | 0.06 (0.33)[**] | 0.06 (0.08) | 21.40 (149.84) | 103.00 (94.68) | 7.13 (13.24) |
| | 6.94 | 0.11 (0.19) | 0.12 (0.32)[*] | 0.06 (0.11) | 24.44 (61.68) | 105.53 (261.13) | 10.89 (23.61) |
| | 8.33 | 0.11 (0.35) | 0.27 (0.51) | 0.08 (0.25) | 21.80 (58.10) | 89.97 (235.23) | 9.50 (22.12) |
| | Max | 0.21 (0.35) | 0.72 (0.50) | 0.12 (0.22) | 33.48 (66.73) | 109.06 (154.86) | 14.07 (52.92) |
| | p | 0.426 | 0.001 | 0.228 | 0.547 | 0.949 | 0.311 |
| | $\eta^2$ | 0.035 | 0.346 | 0.041 | 0.040 | 0.000 | 0.067 |

Notes.
[*],[**] and ** different from maximum speed for a $p < 0.05$ and $<0.01$ (respectively).
Interquartile range (IQR) are presented in parenthesis. The italic values indicate the statistical results.

Table 5 Correlation coefficients between the integrated electromyography (iEMG) of rectus abdominis and erector spinae and propulsion speed.

| | | P-RMS | R-RMS | RE-RMS | P-Int | R-Int | RE-Int |
|---|---|---|---|---|---|---|---|
| Rectus abdominis | r | 0.577 | 0.367 | 0.680 | 0.352 | 0.540 | 0.714 |
| | p | 0.000[**] | 0.010[*] | 0.000[**] | 0.014[*] | 0.000[**] | 0.000[**] |
| Erector spinae | r | 0.232 | 0.551 | 0.273 | 0.097 | 0.083 | 0.253 |
| | p | 0.113 | 0.000[**] | 0.060 | 0.513 | 0.573 | 0.083 |

Notes.
[*],[**] and ** significant at $p < 0.05$ and 0.01 (respectively).

The results confirm our hypothesis that, compared to shoulder, elbow, and wrist joints, trunk movement has a greater influence on wheelchair propulsion speed, while changes in propulsion speed are associated with variations in EMG activity within the trunk muscles. The athlete's trunk exhibits a gradual increase in range of motion and angular velocity with the acceleration of the wheelchair.

Previous research has demonstrated that athletes with lower levels of disability and higher degrees of trunk functionality exhibit a wider range of motion in their trunk (*Gehlsen, Davis & Bahamonde, 1990*; *Goosey-Tolfrey et al., 2001*; *Ridgway, Pope & Wilkerson, 1988*; *Ridgway, Wilkerson & Pope, 1987*). In comparison to those with impaired or absent trunk function, athletes possessing full trunk function are capable of providing stability for the power generated by the shoulder joints and upper limbs (*Gehlsen, Davis & Bahamonde, 1990*), as well as utilizing their trunks to generate greater momentum when pushing on the handrim (*Cooper, 1990*). It may elucidate the disparities in athletic performance among wheelchair racing athletes with varying degrees of disability (*Lewis et al., 2019*). Furthermore, research has demonstrated that inadequate trunk stability heightens the likelihood of shoulder joint injury for wheelchair athletes (*Heyward et al., 2017*; *Yildirim,*

*Comert & Ozengin, 2010*). Additionally, gender, age, and experience are influential factors in trunk movement. Senior and elite male Paralympic participants exhibit greater trunk movements than their female counterparts (*Lewis et al., 2017*) and younger male athletes (*Goosey, Fowler & Campbell, 1997*). Elite male wheelchair athletes are more prone to utilizing larger trunk movements for acceleration (*Wang et al., 1995*). The dissimilarity in trunk movement among different genders and ages may be attributed to varying levels of strength and experience, with elite male athletes typically possessing superior trunk strength and competition experience. It is noteworthy that a study on the correlation between upper arm strength and sprint performance in wheelchair athletes found no significant correlation between upper arm strength and sprint ability at 40 and 100 meters (*Hoffman et al., 1994*). In the aforementioned study, although the trunk strength of the athletes was not measured, it is plausible that disparities in both trunk disability levels and strength levels among participants may be associated with their short-distance sprinting ability based on the findings and inference research (*Hoffman et al., 1994*).

Based on the above findings, it is probable that athletes possessing greater levels of trunk strength and range of motion will exhibit athletic advantages during the initial phase of a race as well as other periods characterized by acceleration and sprinting. This advantage is particularly pronounced in short-distance events.

The earliest investigation into the kinematics of wheelchair racing athletes revealed that trunk inclination can enhance power generation through gravitational forces, augment force transmission from the trunk to the handrim, alter the point of application of force on the handrim, and diminish reaction forces at the handrim (*Sanderson & Sommer, 1985*). When the trunk is inclined forward and exhibits a greater range of motion, it imparts its gravitational force onto the handrim through the rapid movement of the upper extremities, resulting in enhanced acceleration during the propulsive phase (*Wang et al., 1995*). The flexion of the trunk optimizes the position (*Moss, Fowler & Goosey-Tolfrey, 2005*) and direction (*Goosey-Tolfrey et al., 2001*) of force exerted by the arms and hands while also increasing contact angle and range between the handle ring and the athlete's hands (*Chow et al., 2001*; *Gehlsen, Davis & Bahamonde, 1990*; *Goosey-Tolfrey et al., 2001*; *Moss, Fowler & Goosey-Tolfrey, 2005*), resulting in a larger vertical work distance for hand acceleration (*Wang et al., 1995*). The aforementioned findings may account for the outcomes of the current study (Table 2 & Fig. 2). Despite the absence of significant differences in trunk lean angle across various propulsion speeds, an increase in propulsion speed was associated with a corresponding rise in trunk lean angle.

As demonstrated in this study, the multiple linear stepwise regression analysis revealed that the joint motion range and angular speed of the trunk exhibited the highest standardized beta values (Table 3). Furthermore, significant differences were observed in terms of raised angle ($p < 0.05$, $\eta 2 = 0.164$), range of motion ($p > 0.01$, $\eta 2 = 0.573$), and angular velocity ($p > 0.01$, $\eta 2 = 796$) of an athlete's trunk across different propulsion speeds (Table 2 & Fig. 2). The findings suggest that the athletes were able to enhance the propulsion speed of the wheelchair by increasing both the working distance and angular velocity of the trunk which is consistent with previous research (*Goosey & Campbell, 1998a*; *O'Connor, Robertson & Cooper, 1998*; *Wang et al., 1995*). Notably, the $\eta 2$ value and beta

value of angular speed exceeded those of other variables (Tables 2 and 3), indicating that augmenting angular velocity may be a key factor contributing to the increase in propulsion speed. The EMG results of RE-RMS and RE-Int in the rectus abdominis ($r = 0.568$, $r = 0.714$; EMG variables during trunk downward flexion) exhibited the strongest correlation with propulsion speed among all EMG variables examined in this study (Table 5), indicating that activation of the rectus abdominis is critical for enhancing the angular velocity of the trunk. The increased activation of the rectus abdominis during trunk flexion suggests that it accumulates the main power and momentum, leading to an increase in upper limb exertion on the handrim (Wang et al., 1995), and contributing to a higher propulsion speed. Additionally, there was a positive correlation between EMG activity of the erector spinae and propulsion speed during trunk elevation (R-RMS, $r = 0.551$). The EMG findings of the rectus abdominis and erector spinae, as well as the angular velocity of the trunk, indicate the significance of trunk muscle strength in enhancing the propulsion speed of wheelchair racing T54 athletes.

The results of the present study (Table 5) showed that, in comparison with EMG variables of the rectus abdominis, only propulsion speed at R-RMS was correlated with erector spinae, which is consistent with Kumnerddee et al.'s (2018) findings. This study found no correlation between the back muscle group of the trunk and propulsion speed (Kumnerddee et al., 2018). In the present study, the correlation between rectus abdominis EMG activity and propulsion speed at multiple time points suggests that athletes encounter greater resistance when trunk overcoming flexion than when trunk overcoming raising. On one hand, an increase in propulsion speed may also result in a higher reaction force from the handrim during the downward movement of the athlete's trunk (Sanderson & Sommer, 1985), thereby increasing demand for push power. On the other hand, the athlete's trunk will undergo more rapid downward flexion with increasing speed. Furthermore, higher speeds result in increased air drag on the upper body of the athlete on the actual race track, necessitating greater force to maintain the wheelchair moving forward (Forte et al., 2018b). As wheelchair velocity increases, activation of additional muscles is required by the athlete's trunk to counteract air drag during flexion. As a result, the athlete must engage more abdominal muscle fibers to fulfill the aforementioned workload requirements. The primary source of resistance during trunk flexion arises from the athlete's upper body weight and the velocity of trunk movement. However, as the frequency of trunk movement increases and the posterior muscle groups (such as the erector spinae muscles) elongate during downward flexion, storing elastic potential energy for subsequent trunk rise (Chow et al., 2000), the resistance to trunk raising is partially offset by the stored elastic potential energy in the elongated posterior muscle fibers. These findings may account for the outcomes of both the current study and prior studies (Kumnerddee et al., 2018), In contrast to the strong correlation between rectus abdominis activation and propulsion speed at various time points, the posterior trunk musculature exhibits a weak association with propulsion speed.

Excessive elevation of the trunk negatively impacts the competitive performance of wheelchair racing T54 athletes. Firstly, an excessive range of trunk movement (Goosey, Campbell & Fowler, 1998) as well as head and trunk movements that are too fast

(*Jones et al., 1992*) reduce the economy of movement and increase oxygen consumption (*Goosey, Campbell & Fowler, 2000*). Additionally, alterations in body posture can significantly impact air resistance (*Forte et al., 2018b*), a crucial factor as it constitutes 35% of the overall drag force (*Forte et al., 2018a*). Previous studies have indicated that the impact of air resistance gradually amplifies with an increase in propulsion velocity (*Forte et al., 2019b*), air drag accounts for 46% of the total resistance at a propulsion speed of 6.97 m/s. The effective surface area of a wheelchair athlete increases when they adopt an upright position with their trunk raised, as opposed to a competitive position where the trunk is flexed forward (*Barbosa et al., 2016*). Additionally, the power output of wheelchair athletes can be influenced by up to 2% based on their head position (*Barbosa et al., 2016*). According to *Lewis et al.'s (2017)* study findings, optimizing trunk posture for greater aerodynamics can save male athletes 116 s in a 5,000-meter race. As a result, wheelchair racing athletes must be mindful of excessive range of motion and head position when utilizing their trunk to propel the wheelchair in order to optimize movement economy (*Goosey, Campbell & Fowler, 1998*) and decrease air drag.

The current study is not without limitations. The experiment was conducted in a laboratory setting, lacking the effects of wind resistance and track friction that would be present under real-track conditions. It is recommended that future research investigate the kinematics and EMG characteristics of trunk movements among T54 wheelchair racing athletes on actual tracks. On the other hand, the present study includes athletes who adopt both kneeling and sitting postures. However, we have not conducted an analysis of the differences in trunk kinematics and EMG between these two postures. Furthermore, there is currently no evidence to support any potential differences in trunk kinematics or EMG between these two postures. Future studies could provide confirmation on this matter.

## CONCLUSIONS

The role of trunk movements in wheelchair acceleration and maintaining high speed is crucial for T54 wheelchair racing athletes. Compared to arm movements, the propulsion speed of the wheelchair is more significantly influenced by trunk movements. However, excessive raising of the trunk and head should be avoided by athletes due to increased air drag and decreased movement economy. Coaches and athletes should add core strength training for the trunk, with a focus on flexion and extension muscles, into the T54 wheelchair racing athlete's training plan to potentially enhance their performance in competition.

### Funding
This work was supported by the Research on comprehensive scientific research and service of Wheelchair Racing National Training Team (2019-TOKYO2020&001). The funders had no role in study design, data collection and analysis, decision to publish, or preparation of the manuscript.

## Grant Disclosures

The following grant information was disclosed by the authors:

Research on comprehensive scientific research and service of Wheelchair Racing National Training Team: 2019-TOKYO2020&001.

## Competing Interests

The authors declare there are no competing interests.

## Author Contributions

- Wei Guo conceived and designed the experiments, performed the experiments, analyzed the data, prepared figures and/or tables, authored or reviewed drafts of the article, and approved the final draft.
- Qian Liu conceived and designed the experiments, performed the experiments, analyzed the data, prepared figures and/or tables, authored or reviewed drafts of the article, and approved the final draft.
- Peng Huang conceived and designed the experiments, performed the experiments, authored or reviewed drafts of the article, and approved the final draft.
- Dan Wang performed the experiments, authored or reviewed drafts of the article, and approved the final draft.
- Lin Shi performed the experiments, analyzed the data, prepared figures and/or tables, authored or reviewed drafts of the article, and approved the final draft.
- Dong Han conceived and designed the experiments, performed the experiments, authored or reviewed drafts of the article, and approved the final draft.

## Human Ethics

The following information was supplied relating to ethical approvals (*i.e.*, approving body and any reference numbers):

This study was approved by the Ethics Committee of Shanghai University of Sport (102772021RT104).

## Data Availability

The raw measurements are available in the Supplemental Files.

## Supplemental Information

Supplemental information for this article can be found online at http://dx.doi.org/10.7717/peerj.15792#supplemental-information.

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
