# Peer review of "The effects of trunk kinematics and EMG activity of wheelchair racing T54 athletes on wheelchair propulsion speeds"

_PeerJ, doi:10.7717/peerj.15792_

## Round 0.1 · original submission · Major Revisions

Reviews recommend some elaboration on several sections of the manuscript.

·

Basic reporting

The present study has novelty. However, the introduction is not clear about what is pretended to assess. The methods missing some aspects. The results are missing data and the discussion is too long.
The references are adequate and this reviewer just recommended few studies.
This reviewer was not able to evaluate the English. However, it was understandable and clear. Grammar aspects should be reviewed by a fluent speaker.

Before introduction:
L. 44: (p=0.000, r=538; p=0.039, r=298). – I believe 0. Are missing in r.
“P” should be “p” and “p=0.000” should be “p<0.001”


Introduction:
- This reviewer believes that the first paragraph should be 2nd.
- L.66: It is important to firstly describe the stroke cycle. Then, due the drive phase, the authors may relate it with the trunk flection to increase the propulsion.
- L.67: As for economy, check: https://www.tandfonline.com/doi/abs/10.1080/10255842.2018.1502277
- L79: Before that, is important to highlight the classification categories. For example, T51 should only activate upper limbs. T52 should activate shoulder and upper limbs muscles (some cases include pectoral), but no abdominal activity. Based on this study (T54) the athletes are capable of activate the abdominal and spine erectors, so the trunk flection may be better controlled. Why the need for this study?
- L93: This is possible to explain by kinetics and kinematics. A previous published paper reported that 100m race should not be sufficient to reach top speed. https://iopscience.iop.org/article/10.1088/1361-6404/aa6905/meta. Speed will lenarly increase with distance and every variable will play an important role. However, that is not the same in longer distances.
- L98-101: Only in the first meters. At top speed, handrim contact will play an important role in athlete mean top speed. https://www.intechopen.com/chapters/49463
- L102: Still missing the research gap - So far the authors did not explain why is important to evaluate a T54 athletes. Is to provide strategies to improve performance? or to be related with classification issues?

Experimental design

Methods
L113-114: This selection should be understandable in introduction. If I’m studying classification my question will be: "If they are able to perform trunk flection why do this study?" But, if I’m concerned about performance the question will be: "How much can trunk flection affect performance?". The direction should be clear in introduction. However, based on results (abstract) I believe that you were concerned with performance.
L126: The wheelchair will play an important role in sitting height, so it is important to present the used wheelchairs information’s. Additionally, how individual wheelchairs may affect the performance? Are all the wheelchair adequate to each participant? What about small shifts? This should be included in limitations because the authors did not assessed efficiency.
L171-172: please consider this references:
Fuss FK. 2009. Influence of mass on the speed of wheelchair racing. Sports Eng. 12:41–53.
https://www.intechopen.com/chapters/49463


L190-195: Missing effect sizes (eta squared) for ANOVA and (cohen d) for T-Test. Also you should include in results. References will be required.

Validity of the findings

Results
It is missing decriptive data of EMG. such as RMS, Peaks, means, lower values, etc... Please provide figures or tables with descriptive data.
Table 2. Speed should be in m/s. Change accordingly in the tables and text. p values: sometimes "p" appear "P" - attention, should be "p". Revise in the manuscript.
Figure 2: What is a raised angle and a lean angle? It is not clear in methods or presented in introduction. Neither here. What this visually represents in the wheelchair athlete kinematics? What’s the position in a raised angle and in a lean angle? -> this is also missing in discussion.
Check “p values” in tables as presented before.
L.338-343 is not necessary. It was already said.
L419-423: Also not necessary, already said that.
L427: “range of trunk movements reasonable”. What’s a range of trunk movements reasonable? Not explained in introduction or discussion. Please check some of the recommended references support this in discussion.

Additional comments

Nothing more to report.

Reviewer 2 ·

Basic reporting

Thank you for the opportunity to review manuscript ‘The effects of trunk kinematics and EMG activity ofT54 wheelchair racing athletes on wheelchair propulsion speeds. This manuscript explore the influence of the trunk kinematic characteristics and trunk muscle EMG activity of wheelchair racing athletes on different racing wheelchair propulsion speeds. The authors showed an interesting point about the movement of the athlete's trunk that has the largest effect on wheelchair propulsion speed and running speed to avoid increased air drag caused by the excessive trunk raising. However, there are some general and major comments that need to be corrected. I hope the comments are found helpful to the authors for improving their manuscript.

General comments

1. The apparatus, analytical methods, and statistical analysis used in the study were quite appropriate in terms of accuracy and procedures, and thus the obtained data were highly reliable. In addition, the analysis items are appropriate based on the results of previous studies, and the conclusions drawn are considered to be beneficial for scientist and coaches.
2. The manuscript provides a thorough background of previous research from the literature to position the need and purpose for the current study. The authors provide a design to examine the trunk kinematics and EMG activity of wheelchair propulsion speeds, however, the experimental design must be properly explained, especially with regarding to data collection and data processing.
3. Please note that your manuscript would highly benefit from language editing to improve the scientific language.

Major comments
abstract
1. In my opinion, method in abstract could be more specific, provides a clear statistic framework of the paper.
2. In the results and conclusion part, the author should be more clearly outlines the study put forward, and highlights the important aspects of the study with some practical implications.
3. What do authors mean “ to avoid increased air drag caused by the excessive trunk raising? Do authors have any evidence from the current study that the range of movement of athletes’ trunk at different speed have directly affected to increased air drag?

introduction
1. Line 52-65: I think part makes perfect sense by introducing the topic. Yet, if information about the speed at which a wheelchair propels itself is taken into account, it should also make sense to include information about arm and trunk strength as well as together with the information regarding trunk and arm movement.
2. Line 98-101, please add references to support the idea.
3. What about the correlations hypotheses?
4. Introduction can be improved. A better contextualization in the introduction referring to the latest evidence on the EMG activity of the upper limb muscles, kinematics of trunk and arms propulsion and a clearly summarize the current state of the topic would strengthen the manuscript.

Experimental design

Methods

1. The information on athlete’s characteristics, training/competition experience and inclusion criteria was excellently explained by the authors. What about exclusion criteria?
2. Line 115, A total of twelve athletes ….. please insert the mean (SD) of the age, weight and, sitting height in order to make a better characterization of the sample.
3. The authors did a great job explaining about apparatus and instrumentation. The authors provide an in-depth experimental design to examine the kinematic and EMG characteristics of the trunk movement.
4. How about marker reconstruction accuracy and calibrated volume from the kinematic data collection?
5. Line 136-137, EMG electrodes were placed in the rectus abdominis and erector spinus according to the positions suggested in the ABC of EMG. How about the location of electrodes to SENIAM recommendations ?
6. Line 149, please explain in detail the testing protocol. How long for the experimental protocol in each speed?
7. Line 149, what do authors mean “three relevant records for each speed” ?
8. Line 154-155, why do authors testing a maximum voluntary contraction test (MVC) after data collection? Why do not test before data collection? If this protocol was already used by someone, please add a reference here.
9. Please explain how did the author determined maximal voluntary contraction for each muscle? And how was the protocol used?
10. Line 161-165, if this protocol was already used by someone, please add a reference here.

Statistical

1. Line 188-195, in my opinion, the statistics section could be more precise, and that a relationship with the previous sentence and the information that follows would make this easier to read and understand.
2. Line 190-192, please review and revise the sentence.
3. When the author choose to analyse the data using a multiple regression, a critical part of the process involves checking to make sure that the data needs to show homoscedasticity of residuals and must not show multicollinearity. Have the authors justified their assumptions?

Validity of the findings

Results

1. Line 207-213, please provided the estimates of effect size values (partial eta squared (η2) or omega squared (ω2))
2. Line 208, a small point but rise is spelt wrongly raised
3. Line 226, please delete :
4. In the table 2, it would be great providing the p-values for anyone who might want to include your results in a meta-analysis sometime in the future.
5. In my opinion, the manuscript has a lot of tables that confuses the reader. Please think about eliminating some or grouping the most crucial items together in one table.
6. In the table 6, you mentioned and report the integrated EMG values. Please explain how did the author determined and analyzing iEMG particularly in data processing part (line 161-180)

Discussion and Conclusion

1. In the first paragraph of discussion, after the main objective, the reader could better understand and follow discussion if some general conclusions and logically explain the findings are provided, or even, an answer to the hypothesis. This is a suggestion, but it depends on the individual opinion of the authors.
2. Considering the great number of kinematic and EMG variables analysed, the authors could do a greater effort in the results interpretation, discussion and conclusion. Specially, discussion section could be considerably improved according to the following points:
- This section would benefit in some parts from synthetizing the results of the trunk kinematics and EMG activity. In the present format (a different subsection for each propulsion speeds), it lacks a connection between results of the trunk kinematic variables, trunk muscles EMG activity, and trunk strength effect on wheelchair propulsion speeds. If some trunk kinematic and trunk muscles EMG activity parameters are influence and predictors of the wheelchair propulsion speeds, this should be clearly highlighted in discussion.
- In general, section could benefit from a reduction in size and from avoiding some statements too vague.
- There is a point that should be clarified by authors: the T54 wheelchair racing class in the current study comprises athletes with kneeling posture (polio, spinal cord injury and, amputee) and seated which is comprises athletes with normal arm muscle strength and trunk strength ranging from partial to normal. Therefore, discussion should provide comprehensive information particularly trunk kinematic and trunk muscles EMG which is meaningful interpretation and, in this regard, are an exemplar for future studies.
- What is the interest of saying that “The trunk movements of T54 wheelchair racing athletes play an important role in both wheelchair acceleration and speed; increasing the movement range and speed of the trunk can significantly improve the propulsion speed of the racing wheelchair”. Maybe the wheelchair propulsion speeds are indeed associated with wheelchair–athlete interface particularly the trunk and arm strength.
- Some limitations were not pointed out and recognized by the authors in their response. This is important to be included in the manuscript.

---

## Round 0.2 · Minor Revisions

Both reviewers are happy and noted that the manuscript has improved. However, there are some minor issues that deserve the authors' best attention.

·

Basic reporting

No comment.

Experimental design

no comment

Validity of the findings

no comment

Additional comments

no comment

Reviewer 2 ·

Basic reporting

I would like to thank the authors for addressing my initial comments. I must commend the authors for their detailed response, informative comments and thoughtful revisions to this article. Following the revision to the article, some of my additional comments relate to some of the amendments made, and the authors may therefore wish to discuss these particular suggestions with the editor.
Basic Reporting
1) Abstract: I believe this part of the article to often be the most important. In the abstract clearly outlines the study put forward, and highlights the important aspects of the study. Some minor revisions for the authors to consider;
(1.1) what’s the main objective of this work between (i) to examine the impact of trunk kinematic characteristics and trunk muscle electromyography (EMG) activity on various propulsion speeds as mentioned in the abstract or (ii) to analyze the kinematics and EMG characteristics of the trunk in T54 wheelchair racers at various propulsion speeds as mention in the introduction part?
(1.2) “Additionally, the Trigno Wireless EMG system was employed to collect synchronous surface electromyography (EMG) data of the rectus abdominis and erector spinae muscles in wheelchair athletes.” Please consider deleted “in wheelchair athletes”.
(1.3) “Two multiple linear stepwise regression models were constructed for each group of propulsion speed, incorporating the variables identified as significant through correlation coefficient tests (1) and (2).” Please consider rewriting this sentence.
(1.4) The sentence “Four of six variables from the EMG of the athlete's rectus abdominis showed differences at different speeds (p<0.01), one of six showed differences in erector spinae (p<0.01). All six variables derived from the EMG signals of the athlete's rectus abdominis exhibited significant correlation with propulsion speed (p<0.05, r>0.3), while one variable obtained from erector spinae was found to be significantly correlated with wheelchair propulsion speed (p<0.01, r=0.551)” is hard to understand. Please consider rewriting this sentence.
(1.5) Please present the main results in a more concise manner.
2) Introduction:
(2.1) Line 102-105, 105-110, 110-113, 132-134 please consider rewriting this sentence by focus on the idea, not on the authors.
(2.2) Line 150-151: “Trunk movements vary across different age groups and genders of wheelchair racing athletes” How about the performance tiers?
(2.3) Line 152: youthful or young?
(2.4) Line 159-160: “Nevertheless, scant research has been conducted thus far regarding the kinematics and EMG activity of T54 athletes' trunk”. Please consider rewriting this sentence.
(2.5) Line 182-186: As I mentioned in the previous, comments please consider writing the sentence by focus on the idea, not on the authors. This is a suggestion, but it depends on the individual opinion of the authors.
(2.6) Line 205: Please clarify the “timing parameters”.
(2.7) English grammar requires improvement throughout the paper.

Experimental design

3) Method: Overall, this manuscript is generally well explained and written in the instrumentation, procedure, data collection and data processing. Some minor revisions for the authors to consider;
(3.1) Line 300-302: please add a reference at the end of the sentence.
(3.2) Line 352: Statistical analysis?
(3.3) Line 36: “Y-axis represents joint extension or adduction in the coronal plane” or “Y-axis represents joint abduction or adduction in the coronal plane”.
(3.4) Line 383: “Test results with a p value < 0.05 were deemed statistically significant” Please consider rewriting this sentence.
(3.5) For the EMG data processing part, please add the explanation the iEMG work as present in the “Metric Intergrate” of Visual3D
(3.6) It is a challenge to the authors to use parametric and non-parametric in a paper, the authors should consider providing a constructive explained, especially with regarding to see results from multiple analyses - helps to confirm findings or show that findings vary under certain conditions.

Validity of the findings

(4) Results: As I mentioned in the previous comments that I would strongly advise the author to rewrite their results to produce a more contextualized introduction and highlights the important results. The results part is now a much better systematic and reflection of the objectives and findings of the research. Some minor revisions for the authors to consider;
(4.1) The table 1 can be deleted. This is a suggestion, but it depends on the individual opinion of the authors.
(4.2) Table 2: please considered rewrite to;
*and** different from maximum speed for a p < 0.05 and <0.01 (respectively).
#and## different from speed at 8.33 m/s for a p < 0.05 and <0.01 (respectively).
(4.3) Table 4: Please consider rewriting this the heading of table 4 “The integrated electromyography (iEMG), median and interquartile range (IQR) of rectus abdominis and erector spinae at different propulsion speeds”
*and** different from maximum speed for a p < 0.05 and <0.01 (respectively).
(4.4) Table 5: Correlation coefficients between the integrated electromyography (iEMG) of rectus abdominis and erector spinae and propulsion speed.
* and ** significant at p < 0.05 and 0.001 (respectively).ctively).
(5) Discussion and Conclusion
(5.1) Line 542-548, this paragraph is extremely important in terms of letting the reader know what the summarize of the study, general conclusions and logically explain the findings are provided, or even, an answer to the hypothesis. The discussion part is now a much better systematic and reflection findings of the research.
(5.2) Line 805-811: Overall, finding provide strong material for the use in sequencing of improving propulsion speeds of the T54 wheelchair racing athlete’s strategy. It was the relevant in this paragraph showing the complementary trends in the integrated of biomechanics and training perspective. This would lend support to the validity of your result. However, I would have wished to see more information on the practical application on the specific core and upper training method for improving propulsion speed ?

Additional comments

English grammar requires improvement throughout the paper

Annotated reviews are not available for download in order to protect the identity of reviewers who chose to remain anonymous.

---

## Round 0.3 · accepted · Accept

The authors have provided a detailed and thorough response to the comments addressed by the reviewer.

Reviewer 2 ·

Basic reporting

The authors have provided a nicely detailed and thorough response to the comments from the previous review and have addressed my major concerns. The new version of manuscript provides a thorough background of previous research from the literature to position the need and purpose for the current study. The authors also provide an in-depth experimental design to examine the objectives. Given the complexity involved, the author has produced many positive and welcome outcomes in the results and discussion parts.
However, one response needs additional details and I have only one recommendation for the authors to consider:
How about include Acknowledgements ?

The paper is satisfactory and suitable for publication.

Experimental design

-

Validity of the findings

-

Additional comments

-